# Effect of sea-salt aerosol on tropospheric bromine chemistry

Lei Zhu[1], Daniel J. Jacob[1,2], Sebastian D. Eastham[3], Melissa P. Sulprizio[1], Xuan Wang[1], Tomás Sherwen[4,5], Mat J. Evans[4,5], Qianjie Chen[6,a], Becky Alexander[6], Theodore K. Koenig[7,8], Rainer Volkamer[7,8], L. Gregory Huey[9], Michael Le Breton[10,11], Thomas J Bannan[10], and Carl J. Percival[10,b]

[1]John A. Paulson School of Engineering and Applied Sciences, Harvard University, Cambridge, MA, USA
[2]Department of Earth and Planetary Sciences, Harvard University, Cambridge, MA, USA
[3]Laboratory for Aviation and the Environment, Massachusetts Institute of Technology, Cambridge, MA, USA
[4]Wolfson Atmospheric Chemistry Laboratories, Department of Chemistry, University of York, York, UK
[5]National Centre for Atmospheric Science (NCAS), University of York, York, UK
[6]Department of Atmospheric Sciences, University of Washington, Seattle, WA, USA
[7]Department of Chemistry, University of Colorado, Boulder, CO, USA
[8]Cooperative Institute for Research in Environmental Sciences (CIRES), Boulder, CO, USA
[9]School of Earth & Atmospheric Sciences, Georgia Tech, Atlanta, Georgia, USA
[10]The Centre for Atmospheric Science, School of Earth, Atmospheric and Environmental Sciences, University of Manchester, Simon Building, Brunswick Street, Manchester, M13 9PL, UK
[11]Department of Chemistry and Molecular Biology, University of Gothenburg, Medicinaregatan 9 C, 40530, Gothenburg, Sweden
[a]now at: Department of Chemistry, University of Michigan, Ann Arbor, MI, USA
[b]now at: Jet Propulsion Laboratory, California Institute of Technology, 4800 Oak Grove Drive, Pasadena, CA, USA

*Correspondence to*: Lei Zhu (leizhu@fas.harvard.edu)

**Abstract.** Bromine radicals influence global tropospheric chemistry by depleting ozone and by oxidizing elemental mercury and reduced sulfur species. Observations typically indicate a 50% depletion of sea salt aerosol (SSA) bromide relative to seawater composition, implying that SSA debromination could be the dominant global source of tropospheric bromine. However, it has been difficult to reconcile this large source with the relatively low bromine monoxide (BrO) mixing ratios observed in the marine boundary layer (MBL). Here we present a new mechanistic description of SSA debromination in the GEOS-Chem global atmospheric chemistry model with a detailed representation of halogen (Cl, Br, and I) chemistry. We show that observed levels of SSA debromination can be reproduced in a manner consistent with observed BrO mixing ratios. Bromine radical sinks from the HOBr + S(IV) heterogeneous reactions and from ocean emission of acetaldehyde are critical in moderating tropospheric BrO levels. The resulting HBr is rapidly taken up by SSA and also deposited. Observations of SSA debromination at southern mid-latitudes in summer suggest that model uptake of HBr by SSA may be too fast. The model provides a successful simulation of free tropospheric BrO in the tropics and mid-latitudes summer, where the bromine radical sink from the HOBr + S(IV) reactions is compensated by more efficient HOBr-driven recycling in clouds compared to previous GEOS-Chem versions. Simulated BrO in the MBL is generally much higher in winter than in summer due to a combination of greater SSA emission and slower conversion of bromine radicals to HBr. An outstanding issue in the model is the overestimate of free tropospheric BrO in extratropical winter-spring, possibly reflecting an overestimate of the HOBr/HBr ratio under these conditions where the dominant HOBr source is hydrolysis of $BrNO_3$.

## 1 Introduction

Bromine radicals ($BrO_x \equiv Br + BrO$) influence global tropospheric chemistry by depleting ozone and thus OH, and by oxidizing species such as elemental mercury and dimethylsulfide (Saiz-Lopez and von Glasow, 2012; Simpson et al., 2015; Long et al., 2014). Tropospheric bromine radical chemistry is initiated by the production of reactive inorganic bromine ($Br_y$) from sea salt aerosol (SSA) debromination, decomposition of organobromines primarily of marine origin ($CHBr_3$, $CH_2Br_2$, and $CH_3Br$), and transport from the stratosphere. Within the $Br_y$ family, bromine radicals cycle with non-radical reservoir species such as HBr, HOBr, $BrNO_2$, $BrNO_3$, $Br_2$, BrCl, and IBr. Loss of $Br_y$ is by wet and dry deposition to the surface, mainly as HBr which is highly soluble in water.

Sea salt aerosol (SSA) is thought to be the largest source of tropospheric bromine. Observations show extensive debromination of SSA relative to seawater composition (Sander et al., 2003; Newberg et al., 2005). Parrella et al. (2012) estimate a global $Br_y$ source of 1420 Gg Br $a^{-1}$ from SSA debromination, as compared with 520 Gg $a^{-1}$ from organobromines and 36 Gg $a^{-1}$ from the stratosphere. Volatilization of bromide from SSA can take place by heterogeneous reactions with HOBr, HOCl, $N_2O_5$, ozone, and $ClNO_3$ (Vogt et al., 1996; Hirokawa et al., 1998; Keene et al., 1998; Firckert et al., 1999; von Glasow et al., 2002a, b; Yang et al., 2005; Ordóñez et al., 2012; Saiz-Lopez et al., 2012; Long et al., 2014). A standing conundrum has been that observations of BrO in the marine boundary layer (MBL) do not show large enhancements relative to the free troposphere, where background mixing ratios are typically of the order of 1 ppt (ppt $\equiv$ pmol $mol^{-1}$, Leser et al., 2003; Sander et al., 2003; Theys et al., 2011; Volkamer et al. 2015; Wang et al., 2015; Le Breton et al., 2017). Ozone observations in the MBL similarly do not show depletion that would be expected from high mixing ratios of BrO (de Laat and Lelieveld, 2000; Sherwen et al., 2016). This has led recent global models not to include SSA debromination as a source of $Br_y$ (Schmidt et al., 2016; Sherwen et al., 2016).

Here we present a new mechanistic description of sea salt debromination in the GEOS-Chem global 3-D model of tropospheric chemistry including detailed representation of halogens (Cl, Br, and I) (Sherwen et al., 2016; Wang et al., 2019). We find that we can reproduce the observed levels of SSA debromination while also being consistent with the relatively low BrO mixing ratios observed in the MBL. This is because the previously-recognized fast production of $Br_y$ from SSA debromination is offset by fast removal Br atoms by acetaldehyde emitted from the ocean (Toyota et al., 2004; Millet et al., 2010; Badia et al., 2019) and by fast removal of HOBr by dissolved $SO_2$ (S(IV)) in cloud (Long et al., 2014; Chen et al., 2017). We examine the implications for the global budget of tropospheric bromine and tropospheric oxidants.

## 2 Data and methods

We use GEOS-Chem 12.3.0 (https://doi.org/10.5281/zenodo.2620535), which includes a detailed representation of ozone-$NO_x$-VOC-aerosol-halogen tropospheric chemistry (Sherwen et al., 2016), to which we have added the comprehensive

tropospheric chlorine chemistry of Wang et al. (2019) with some additional modifications as described below. The model is driven by assimilated meteorological data for 2011–2012 from the Modern-Era Retrospective analysis for Research and Applications, Version 2 (MERRA2) produced by the NASA Global Modeling and Assimilation Office (Gelaro et al., 2017). The horizontal resolution of MERRA2 is $0.5° \times 0.625°$, and is degraded here to $4° \times 5°$ for input to GEOS-Chem. Dynamic

and chemical time steps are 30 and 60 minutes, respectively. GEOS-Chem stratospheric chemistry (Eastham et al., 2014) is linearized following Murray et al. (2012) to serve as boundary conditions for stratospheric input to the troposphere. The model is spun up for 1 year and we use simulation results for 2012.

Tropospheric bromine chemistry in GEOS-Chem was first introduced by Parrella et al. (2012). Loss of $Br_y$ from the troposphere

is mainly by deposition of HBr, which is highly water-soluble unlike HOBr or BrO. Parrella et al. (2012) found that the acid-catalyzed $HOBr + Br^-$ reaction taking place in liquid-water clouds, ice crystal quasi-liquid surfaces, and aqueous aerosols was critical for recycling bromide and maintaining background tropospheric BrO at observed ~ 1 ppt levels:

$$HOBr(aq) + Br^- + H^+ \rightarrow Br_2(g) + H_2O \tag{R1}$$

$$Br_2 + hv \rightarrow 2\ Br \tag{R2}$$

$$Br + O_3 \rightarrow BrO + O_2 \tag{R3}$$

The current standard version of the model (GEOS-Chem 12.3.0) includes more extensive heterogeneous bromine chemistry (Schmidt et al., 2016), coupling to other halogens (Sherwen et al. 2016), $HOBr + S(IV)$ reactions in clouds (Chen et al., 2017), and oceanic emission of acetaldehyde which reacts rapidly with Br atoms (Toyota et al., 2004; Millet et al., 2010; Badia et al., 2019). Wang et al. (2019) more recently added a comprehensive treatment of tropospheric chlorine chemistry in GEOS-Chem

including explicit accounting of SSA chloride volatilization and aerosol pH. The model does not attempt to simulate the fast but localized bromine chemistry taking place in the Arctic MBL in spring due to volatilization of bromine deposited on sea ice (Simpson et al., 2015).

Here we added several updates to the computation of heterogeneous chemistry recycling bromine radicals. Uptake of HOBr(aq)

by clouds involves competition between reactions with $Br^-$, $Cl^-$, $HSO_3^-$, and $SO_3^{2-}$, as given by (R1), (R4), and (R5):

$$HOBr(aq) + Cl^- + H^+ \rightarrow BrCl(g) + H_2O \tag{R4}$$

$$HOBr(aq) + HSO_3^- \rightarrow H_2O + BrSO_3^- \tag{R5a}$$

$$HOBr(aq) + SO_3^{2-} \rightarrow OH^- + BrSO_3^- \tag{R5b}$$

$$BrSO_3^- + H_2O \rightarrow SO_4^{2-} + Br^- + 2H^+ \tag{R5c}$$

(R1) and (R4) recycle bromine radicals while (R5) is effectively a terminal sink because of HBr deposition. The standard GEOS-Chem code computes the reactive uptake coefficient ($\gamma$) independently for each pathway, which incorrectly assumes that other pathways do not affect mass transfer. Here we express the first-order aqueous-phase loss of HOBr as a sum of the four pathways to compute $\gamma$, and we then distribute the loss by pathways on the basis of the relative rates (see Supplemental

Materials Text S1). When calculating HOBr uptake by ice crystals, we assume a radius of 38.5 μm based on cloud observations (Fu, 1996) rather than 75 μm in the standard GEOS-Chem code, and further increase the effective surface area of ice crystals by a factor of 10 to account for their irregular shape (Schmitt and Heymsfield, 2005). Finally, we correct a registration error for SSA alkalinity in the standard GEOS-Chem code that caused underestimate of alkalinity titration (see Supplemental

Materials Text S2). The overall result of our updates is to have more efficient heterogeneous recycling of bromine radicals, both in the MBL and in the free troposphere.

The GEOS-Chem SSA simulation is from Jaeglé et al. (2011), who showed that it could reproduce successfully SSA observations over the oceans from ship cruises and coastal/island stations, as well as observations of aerosol optical depth

(AOD) from the AERONET network and the MODIS satellite instrument. The model separates fine ($\leq 0.5$ μm radius) and coarse SSA as two separate transported species. The global dry SSA source in our simulation is 3140 Tg $a^{-1}$. We emit bromide as part of fine and coarse SSA with a seawater ratio of $2.11\times10^{-3}$ kg Br per kg dry SSA (Sander et al., 2003; Lewis and Schwartz, 2004), since observations show that fresh SSA has a bromide content equals that of seawater (Duce and Woodcock, 1971; Duce and Hoffman, 1976; Turekian et al., 2003). SSA alkalinity is emitted at a ratio of 0.07 equivalents per kilogram of

dry SSA, and is depleted by strong acids following Alexander et al. (2005). The cloud pH is calculated as described in Alexander et al. (2012) and is 3.5–6.5 for clean marine conditions, consistent with the observed range (3.8–6.1) reported in the literature (Gioda et al., 2009; Hegg et al., 1984; Lenschow et al., 1988; Vong et al., 1997; Watanabe et al., 2001).

Activation of SSA bromide takes place by heterogeneous reactions with HOBr, ozone, and $ClNO_3$ once alkalinity has been

titrated and SSA is acidified (Hirokawa et al., 1998; Keene et al., 1998; Fickert et al., 1999):

$$Br^- + HOBr(aq) + H^+ \rightarrow Br_2 + H_2O \tag{R1}$$

$$Br^- + O_3(aq) + H^+ \rightarrow HOBr + O_2 \tag{R7}$$

$$Br^- + ClNO_3(aq) + H^+ \rightarrow BrCl + HNO_3 \tag{R8}$$

We also consider parameterized SSA debromination by HOI(aq) following McFiggans et al. (2002), where HOI(aq) may be

taken up from the gas phase or produced by hydrolysis of $INO_2$ and $INO_3$:

$$0.15Br^- + 0.85Cl^- + HOI(aq) + H^+ \rightarrow 0.15IBr + 0.85ICl + H_2O \tag{R9}$$

Inorganic oceanic iodine (HOI and $I_2$) emissions are from Carpenter et al. (2013). Unlike for chloride, SSA debromination does not take place by acid displacement because of the much stronger acidity of HBr than HCl or $HNO_3$ (Sander et al., 2003). On the contrary, uptake of gas-phase HBr can lead to bromine enrichment in SSA.


Sander et al. (2003) introduced the dimensionless enrichment factor (*EF*) as a measure of SSA debromination. *EF* is computed from aerosol measurements as:

$$EF = \frac{([Br^-]/[Na^+])_{measured}}{([Br^-]/[Na^+])_{seawater}} \tag{1}$$

where aerosol [Na$^+$] is assumed to be mainly from sea salt, a reliable assumption in marine air. In GEOS-Chem we treat 'SSA' as a chemically inert tracer, and account for sea-salt bromide as a separate species, therefore *EF* is computed as:

$$EF = \frac{([Br^-]/[SSA])_{SSA\ aerosol}}{([Br^-]/[SSA])_{SSA\ emission}} \tag{2}$$

We sum [Br$^-$] and [SSA] from both fine and coarse SSA to calculate *EF*. We will also present *EF* values for fine and coarse
SSA separately.

## 3 Results and discussion

Figure 1 (top panel) shows the annual mean SSA bromine enhancement factors (*EF)* in surface air computed by GEOS-Chem and compares to annual mean observations compiled by Sander et al. (2003) and from Newberg et al. (2005). We consider 6 island and 4 coastal sites with bulk aerosol *EF* measurements available for more than one year. The observations are for
different years than the GEOS-Chem simulation, but we assume that interannual variability is a minor source of error. The mean GEOS-Chem *EF* averaged over the sites is 0.75 ± 0.23 (± 1 standard deviation), compared with the observed value of 0.66 ± 0.32. SSA bromide over the Southern Ocean in the model is less depleted (*EF* ~ 0.9) than over the northern mid-latitudes (*EF* ~ 0.6), because SSA tends to retain its alkalinity over the Southern Ocean (Alexander et al., 2005; Schmidt et al., 2016; Supplemental Materials Figure S1). Similarly in the observations, mean *EF* is 0.78 ± 0.08 over the Southern Ocean compared
with 0.42 ± 0.11 at northern mid-latitudes. Most of the SSA debromination in the model is from (R1).

SSA mass in GEOS-Chem is dominated by the coarse component over the oceans and by the fine component over land (due to fast dry deposition of the coarse component). Thus the *EF* values for bulk SSA over the ocean are dominated by the coarse component. SSA debromination is more extensive for the fine component of SSA because the initial supply of bromide is less
and loss of alkalinity is more extensive. Some areas of the ocean have weak SSA bromide enrichment (*EF* > 1) because of uptake of HBr. Over land the *EF* values are dominated by the fine SSA component because the coarse component deposits close to the coast. These *EF* values can be very large because Br$_y$ volatilized from the coarse SSA component is then taken up as HBr by the fine SSA after the coarse component has deposited.

The model overestimates the observed *EF* over the Southern Ocean. This appears to reflect a seasonal bias in the model. Figure 2 compares the simulated and observed seasonality at Cape Grim, Tasmania (Ayers et al., 1999; Sander et al., 2003). To our knowledge, this is the only site for which seasonal information is available in the observations. The observed *EF* is 0.6–0.8 for most of the year, consistent with the model, but decreases to below 0.4 in summer while the model does not. The summer minimum in the observations has been attributed to increased SSA acidity (Ayers et al., 1999; Sander et al., 2003). Indeed,
Figure 2 shows that SSA alkalinity in the model is titrated in summer due to the combination of weaker SSA emission (lower winds) and larger photochemical production of strong acids (H$_2$SO$_4$ and HNO$_3$). This drives volatilization of Br$_y$ from SSA,

but we find in the model that the resulting HBr mainly returns to SSA rather than deposits to the surface because SSA emission is still relatively high. Uptake of HBr by SSA proceeds in the model with a reactive uptake coefficient $\gamma = 1.3 \times 10^{-8}$ exp(4290 K / $T$) as recommended by IUPAC (Amman et al., 2013) but with large uncertainty, ranging from –90% to +860% at 278 K.

Figure 3 shows the global budget and speciation of tropospheric $Br_y$ in our simulation. This updates a similar figure by Schmidt et al. (2016) to include SSA debromination, the HOBr + S(IV) reactions in clouds, oceanic emission of acetaldehyde, and full coupling with the other halogens. SSA debromination is the largest global source, mainly from (R1) and (R7) producing $Br_2$ and HOBr respectively. The dominant sink of $Br_y$ is uptake of HBr by SSA, rather than deposition, emphasizing the importance of competition between these two processes in determining the extent of SSA debromination. Bromine radical (Br) is converted

to HBr by formaldehyde, acetaldehyde, $HO_2$ radical, and $\geq C3$ alkenes with relative contributions of 52%, 40%, 7%, and 1%, respectively. In the Schmidt et al. (2016) budget, acetaldehyde contributed only 17% of this bromine radical sink; the larger contribution in our simulation reflects its oceanic emission. Observations from the recent ATom aircraft campaign (Wofsy et al., 2018) over the remote Pacific and Atlantic show mean MBL acetaldehyde mixing ratios within 10% of those simulated by GEOS-Chem including the Millet et al. (2010) ocean source (Bates et al., 2018).

The global tropospheric loading of BrO in our simulation is 8.0 Gg Br, corresponding to a mean tropospheric mixing ratio of 0.86 ppt (1.7 ppt as daytime average). The BrO loading is higher than in previous GEOS-Chem versions starting with 3.8 Gg in Parrella et al. (2012), 5.7 Gg in Schmidt et al. (2016), 6.4 Gg in Sherwen et al. (2016), 3.6 Gg in Chen et al. (2017), and 4.2 Gg in Wang et al. (2019). Wang et al. (2019) described this evolution between versions. Our high BrO loading reflects our

updates to HOBr uptake and correction of SSA alkalinity.

Figure 4 compares simulated tropospheric BrO columns with Global Ozone Monitoring Experiment (GOME)-2 satellite observations from Theys et al. (2011) as a function of season and for different latitudinal bands. Also shown are tropospheric columns from ground-based measurements in Florida, USA (Coburn et al, 2011) and derived from mean aircraft vertical

profiles over the tropical Pacific from the TORERO (Volkamer et al., 2015; Wang et al., 2015; Dix et al., 2016) and CONTRAST (Koenig et al., 2017) campaigns. There is general consistency between these observations. Model results are from our standard simulation and from a sensitivity simulation without SSA debromination. The standard simulation provides a good fit to the observations in the tropics but is much too high at extratropical latitudes in winter and spring. High model mixing ratios under these conditions are due to high SSA emissions and fast bromide recycling via (R1), the latter due to a

large HOBr source from $BrNO_3$ hydrolysis. $BrNO_3$ hydrolysis mainly takes place in cloud (droplets and ice crystals) rather than in aerosols. The dominant global sink for $BrNO_3$ is photolysis (Figure 3), but under extratropical winter-spring conditions we find that hydrolysis is more important because of weak radiation. Another reason for the high modeled BrO in extratropical winter-spring is that removal of $Br_y$ via deposition of HBr becomes slower under these conditions, as described below.

Figure 5 compares simulated vertical profiles with aircraft BrO observations over the tropics in January and February. Schmidt et al. (2016) and Shermen et al. (2016) reported negative biases of 0.6–1.0 ppt in modeled BrO compared with TORERO observations in the upper free troposphere. Our standard simulation shows a much smaller bias (~ 0.3 ppt) because of the impact of faster HOBr uptake on cloud ice (Section 2). Figure 5 also shows BrO results from two sensitivity simulations not including SSA debromination or the HOBr + S(IV) reactions. Model-observation agreement in those two cases is generally not as good as in our standard simulation. As shown in Figure 5, the impact of SSA debromination extends through the depth of the troposphere by increasing BrO by 0.1–0.5 ppt with larger impact near the surface. This is generally seen in other seasons over the tropical latitudes as well (Supplemental Materials Figure S2). For the extratropical latitudes, the impact of SSA debromination on BrO is much larger (up to 3 ppt) especially in winter-spring, which is also reflected in BrO columns (Figure 4). Besides high SSA emissions and fast bromide recycling as described above, we also find that only ~ 15% of $Br_y$ there is present as HBr (Supplemental Materials Figure S3 and S4), reflecting the relatively lower Br/BrO ratio (~ 0.02 at winter while ~ 0.08 at summer) when radiation is weak. $Br_y$ has then a longer lifetime because non-HBr species are much less water-soluble (Parrella et al., 2012) and can be effectively transported to the free troposphere.

Figure 6 shows the simulated global distribution of BrO mixing ratios in surface air in January and July. We attribute the elevated marine surface BrO in winter time to a combination of greater SSA emission and slower removal of $Br_y$ via deposition of HBr. However, we fail to find enough information in previous studies to evaluate the modeled seasonality in marine surface BrO. Only Cape Verde has seasonal information (Read et al., 2008; Mahajan et al., 2010) and shows insignificant seasonal variation consistent with the model, but this is for the tropics. Observations are compiled in Table 1 with corresponding model values. The model is generally consistent with observations in showing daytime mixing ratios in the range 0.5–2 ppt, including low BrO (0.3–0.6) in the MBL measured from aircraft campaigns. The low level of BrO (~ 0.3 ppt) over the eastern tropical Pacific Ocean (Volkamer et al., 2015; Wang et al., 2015; Dix et al., 2016) observed during the TORERO flight campaign is driven by weak SSA emission in the austral summer. The higher BrO in CONTRAST than in CAST is reproduced by the model where it is due to regional variations in SSA emission. The tropical North Atlantic is enhanced with BrO (1–3 ppt) relative to other tropical oceans, both in the model and observations (Tenerife, Cape Verde). This is due in the model to slow removal of $Br_y$ via dry deposition of $BrNO_3$, resulting in elevated $Br_y$ (and BrO through $Br_y$ cycling). We find the production rate of $BrNO_3$ by BrO + $NO_2$ in this region is ~ 76% slower than over the surrounding Atlantic. Isolated hotspots near India (July) and the Caribbean (January and July) in the model correspond to localized hotspots of SSA emissions.

## 4 Conclusions

Observations of sea salt aerosol (SSA) debromination over oceans worldwide imply a large source of bromine radicals in the marine boundary layer (MBL), yet measured BrO mixing ratios in the MBL are relatively low. Here we attempted to reconcile

these observations with a global simulation of tropospheric bromine chemistry in the GEOS-Chem model including detailed representation of processes.

We find that we can successfully simulate the observations of SSA debromination in the literature as measured by the SSA bromine enrichment factor ($EF$). Most of the debromination in the model is by the $HOBr(aq) + Br^- + H^+$ and $O_3(aq) + Br^- + H^+$ reactions taking place in acidified SSA. Debromination is more extensive at northern than southern latitudes because of higher acidity. Observations at southern mid-latitudes show extensive debromination in summer that is not captured by the model and we attribute this to model competition for HBr between uptake by SSA and deposition, where HBr uptake by SSA (taken from the IUPAC recommendation) may be too fast. The model predicts large bromide enrichments for fine SSA over land ($EF > 1$) as bromine lost from the coarse SSA transfers as HBr to the fine SSA that is transported inland.

Our model simulation improves over previous GEOS-Chem versions in the simulation of surface, aircraft, and satellite observations of BrO mixing ratios in the tropics and summertime mid-latitudes. Previous model versions including SSA debromination overestimated BrO in the MBL and underestimated it in the free troposphere. Our lower BrO in the MBL reflects the inclusion of radical sinks to HBr from the $HOBr + S(IV)$ and $CH_3CHO + Br$ reactions. Our higher BrO in the free troposphere reflects more efficient recycling of bromine radicals by HOBr reactions in clouds. However, the model appears to generate excessive free tropospheric BrO in the extratropics in winter-spring. A distinctive feature of these conditions in the model is a HOBr/HBr ratio in excess of unity, reflecting a large source of HOBr from $BrNO_3$ hydrolysis and the inefficient production of HBr in the MBL, allowing MBL bromine to be transported to the free troposphere. Further investigation into the chemistry mechanism and uncertainty may be needed, including uptake of HBr by SSA, uptake of HOBr by aerosols (Roberts et al., 2014), $BrNO_3$ hydrolysis, unexplained observations of oxygenated VOCs in the free troposphere (Volkamer et al., 2015; Andersen et al., 2017; Badia et al., 2019), and oxidation of bromide by ozone involving the $BrOOO^-$ ozonide (Artiglia et al., 2017).

## 5 Acknowledgments

This work was supported by the NSF Atmospheric Chemistry Program. We acknowledge contributions from the TORERO, CONTRAST, and CAST science team. We thank Dexian Chen for the CONTRAST BrO observations. Q.C. and B.A. acknowledge support from NSF AGS 1343077. R.V. and T.K.K. acknowledge funding from NSF AGS-1620530.

## 6 Data availability

The GEOS-Chem model is available at http://acmg.seas.harvard.edu/geos/. GEOS-Chem chemistry mechanism and monthly bromine simulation output used in this study are available at https://doi.org/10.7910/DVN/BADJDE.

## 7 Author contributions

L.Z. and D.J.J. designed the research and wrote the paper; L.Z., D.J.J., S.D.E., M.P.S., X.W., T.S., M.J.E., Q.C., and B.A. led the model development; T.K.K., R.V., L.G.H., M.L., T.J.B., and C.J.P. provided BrO observations.

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

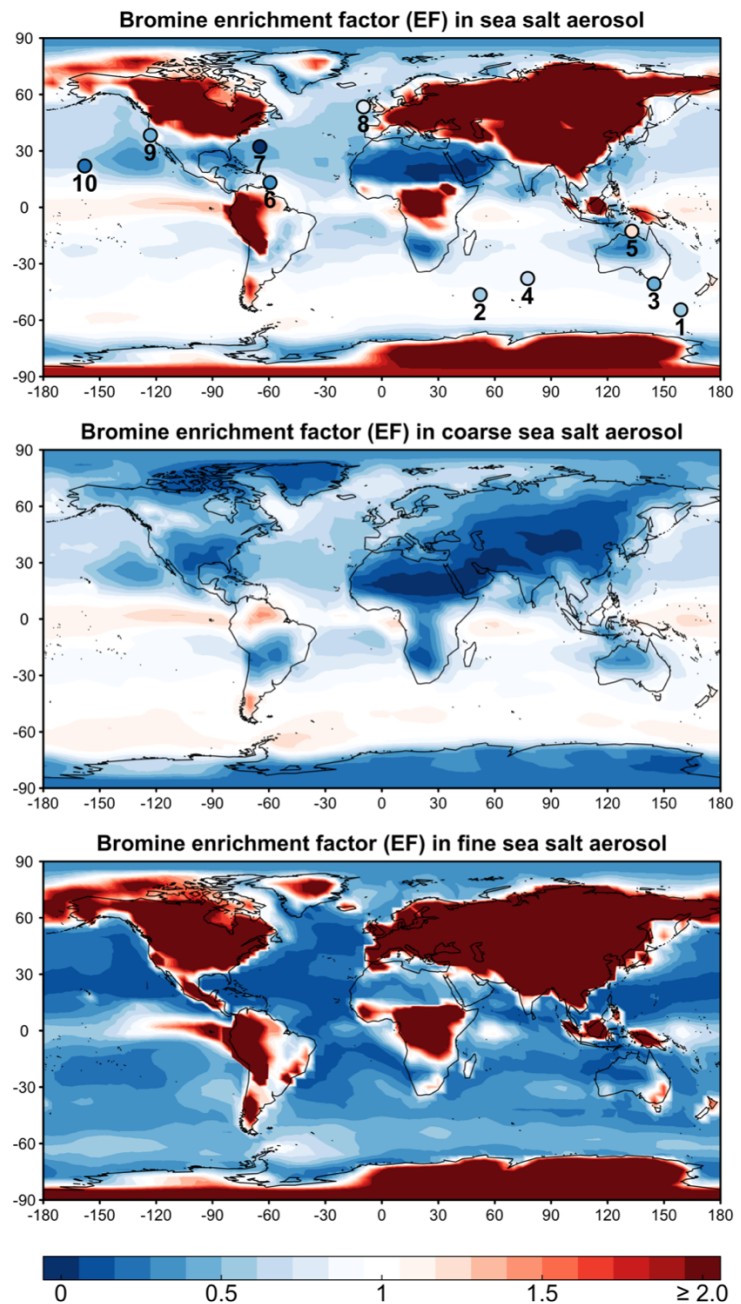

**Figure 1: Annual mean bromine enrichment factor (*EF*) of sea salt aerosol (SSA) in surface air. GEOS-Chem model results (contours) are compared to observations (circles in top panel). The top panel shows results for total SSA (fine + coarse), the middle panel is for coarse SSA (> 0.5 µm radius), and the bottom panel is for fine SSA (≤ 0.5 µm radius). Observations at 10 sites compiled by Sander et al. (2003) and from Newberg et al. (2005) are superimposed as circles and labeled in order 1–10: Macquarie Island, Crozet Islands, Cape Grim, Amsterdam Island, Jabiru, Barbados, Bermuda, Mace Head, Bodega Bay, and Hawaii, respectively. The simulation is for 2012 and the observations are for different years. Color bar saturates at 2.0. Maximum modeled *EF* is 75.0.**

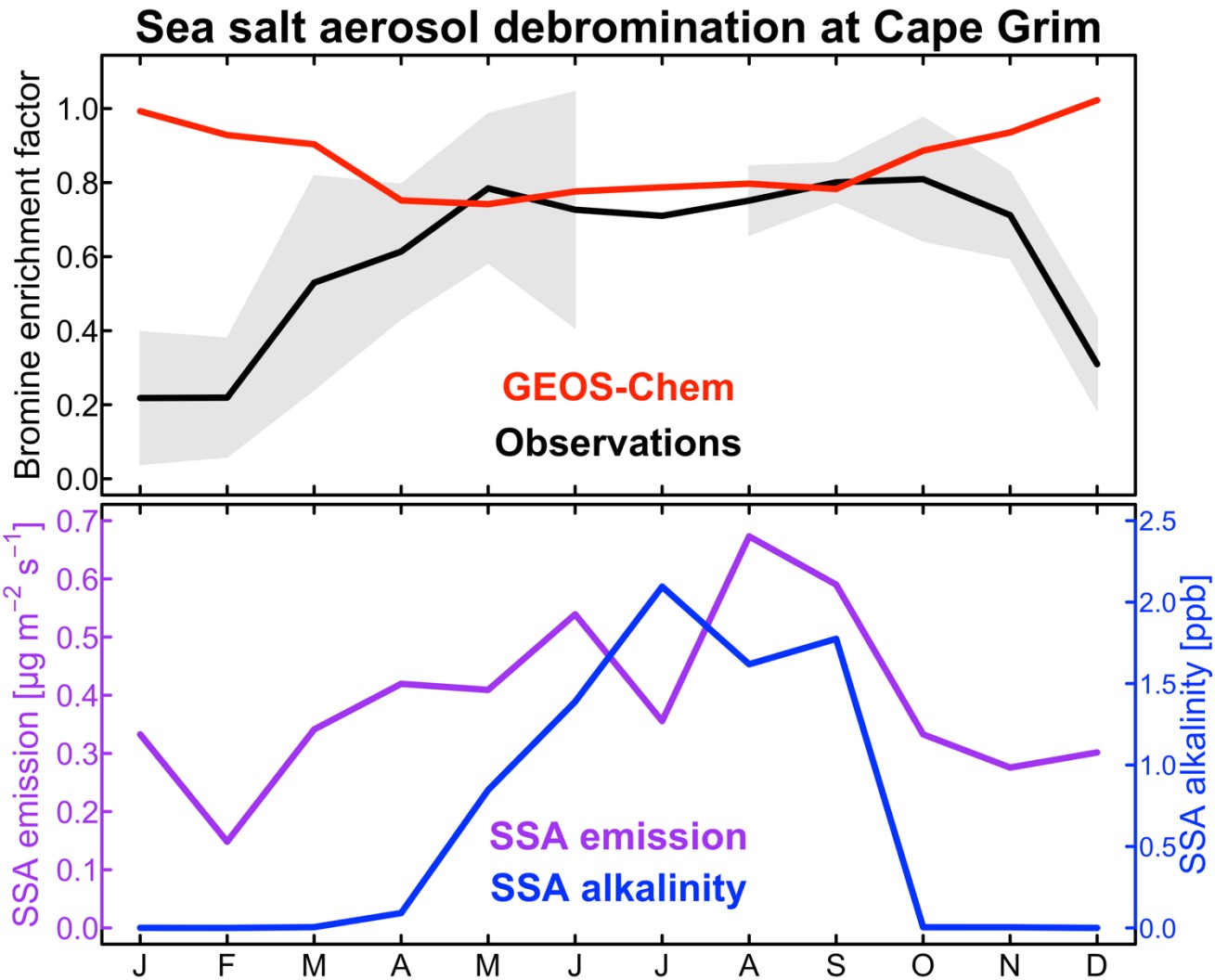

**Figure 2. Seasonal variation of sea salt aerosol (SSA) debromination at Cape Grim, Tasmania (40.7ºS, 144.7ºE). The top panel shows the bromine enrichment factors (*EF*s) of monthly mean SSA in surface air. Observations from Ayers et al. (1999) and Sander et al. (2003) for 1996–1998 are compared to GEOS-Chem model values for 2012. Shading gives the interannual standard deviation in the observations. The bottom panel shows the GEOS-Chem monthly SSA emission flux at Cape Grim (Site 3 in Figure 1) and the SSA alkalinity. The SSA emission flux is for the oceanic fraction of the Cape Grim gridsquare.**

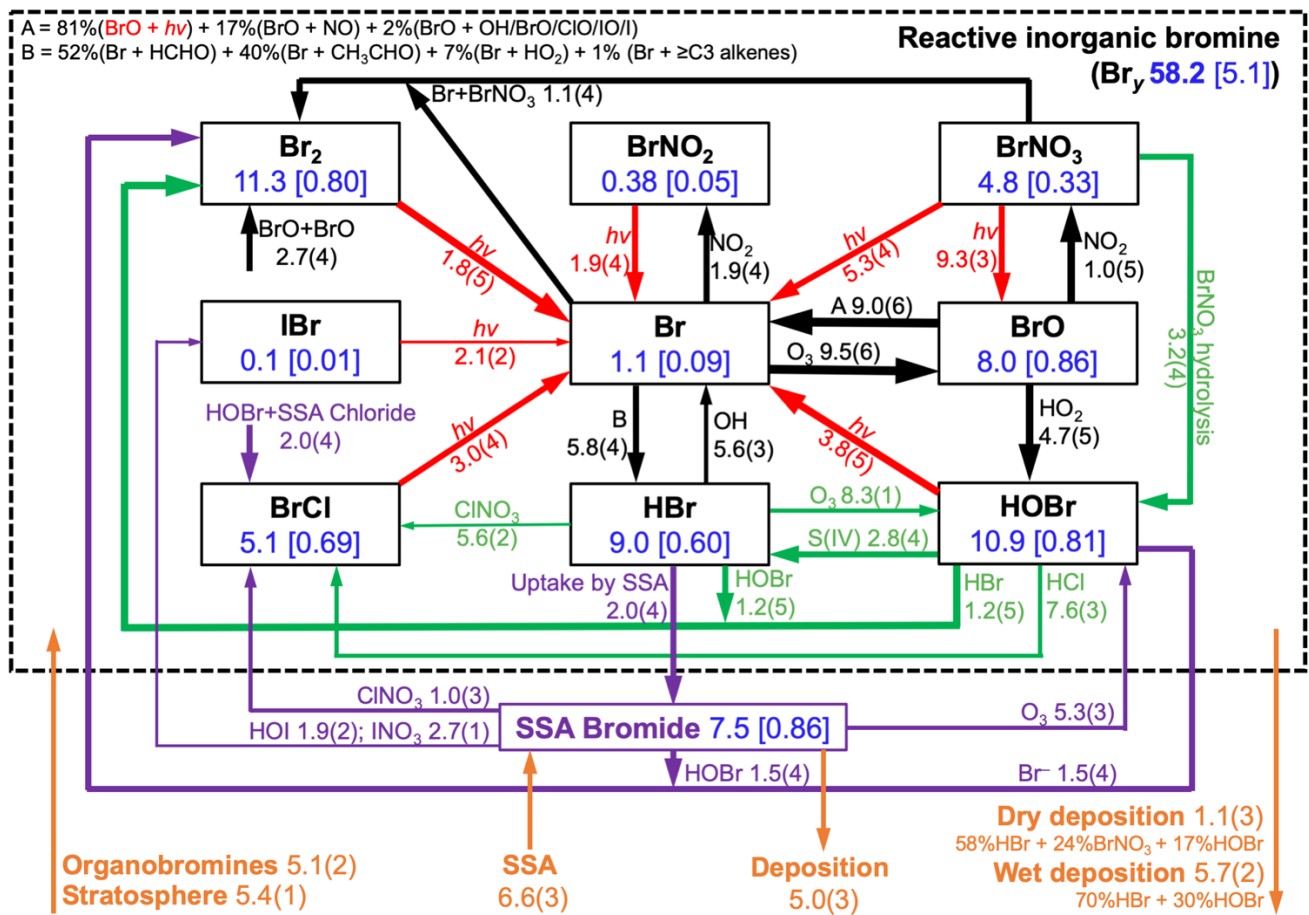

**Figure 3. Global annual mean tropospheric budget and cycling of reactive inorganic bromine (Br$_y$) and sea salt aerosol (SSA) bromide. Results are from our GEOS-Chem simulation for 2012 including SSA debromination. Br$_y$ is defined as the ensemble of species inside the dashed box. Rates are in Gg Br a$^{-1}$, masses in the boxes are in Gg Br, and numbers in brackets are mean mixing ratios (ppt). Read 5.8(4) as 5.8×10$^4$ Gg Br a$^{-1}$. Arrows in black are for gaseous reactions, red for photolysis, purple for heterogeneous reactions in SSA, and green for other heterogeneous reactions taking place in cloud and sulfate aerosol. Sources and sinks of total inorganic bromine (Br$_y$ + SSA bromide) are in orange. Arrow thickness scales with its corresponding rate.**

# Tropospheric BrO column: seasonal variation

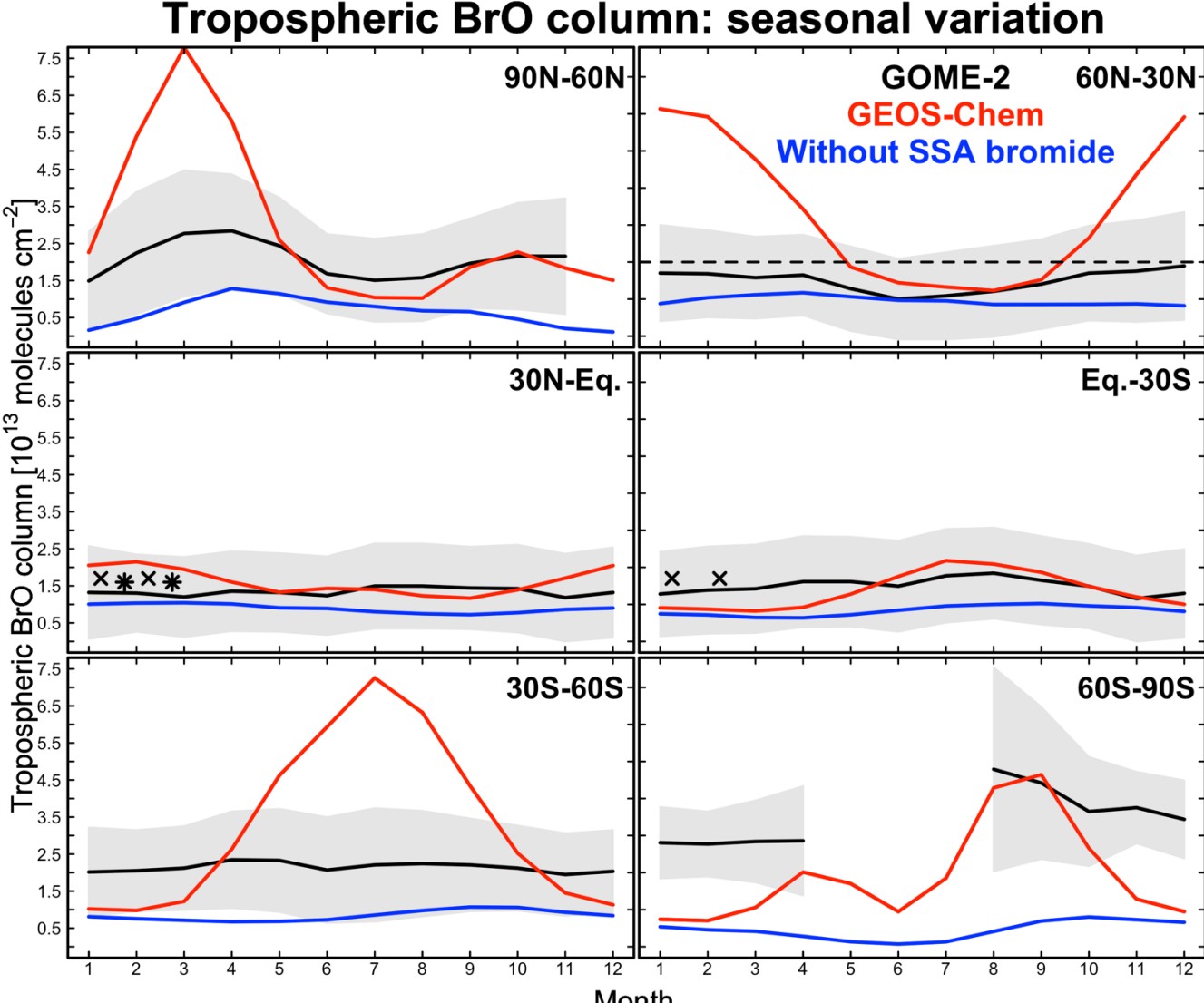

**Figure 4. Seasonal variation of zonal mean tropospheric BrO columns in different latitudinal bands. Monthly GOME-2 BrO observations are for 2007 and taken from Theys et al. (2011); shading represents one standard deviation about the monthly mean GOME-2 BrO columns. GEOS-Chem BrO columns are sampled at the GOME-2 local overpass time (09:00–10:00). Red lines are from our standard simulation including sea salt aerosol (SSA) debromination, blue lines are from a sensitivity simulation without SSA debromination. The black dashed line indicates observations for 2009–2011 reported by Coburn et al. (2011) in Florida, USA without seasonality information. Black crosses and stars represent average BrO columns measured during aircraft campaigns over the eastern tropical Pacific (Volkamer et al., 2015; Wang et al., 2015; Dix et al., 2016) and western tropical Pacific (Koenig et al., 2017), respectively.**

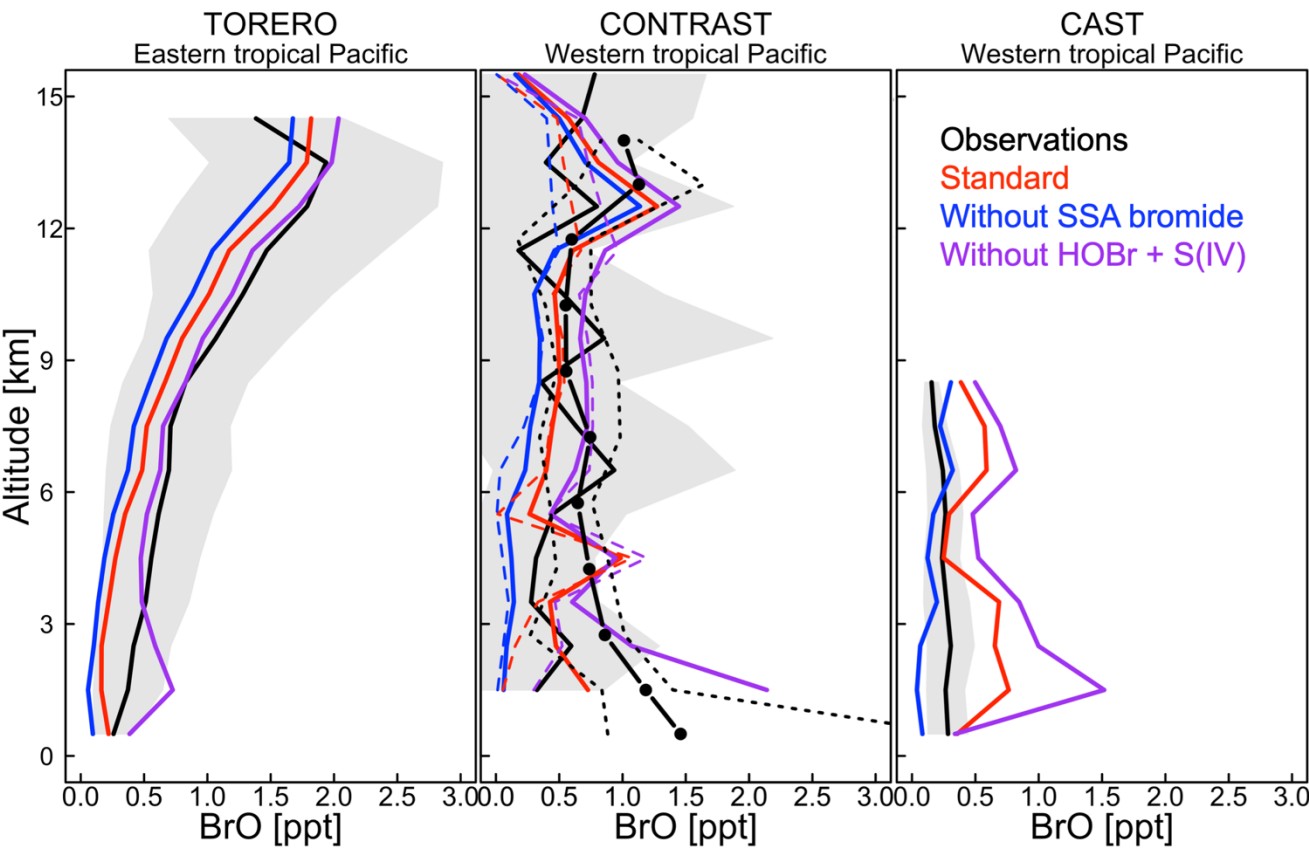

**Figure 5. Vertical profiles of BrO mixing ratios over the tropical Pacific. Observations from the TORERO (Volkamer et al., 2015; Wang et al., 2015; Dix et al., 2016), CONTRAST (Chen et al., 2016; Koenig et al., 2017), and CAST (Le Breton et al., 2017) aircraft campaigns are compared to model values. Solid black lines indicate mean observed values in 1-km vertical bins and with standard deviations (shading). We use two independent CONTRAST BrO data sets. The black line with dots shows the median values (dashed lines are 50% and 75% quantiles) as reported in Koenig et al. (2017). The solid line denotes the mean values from Chen et al. (2016). GEOS-Chem is sampled along the flight tracks at the time of the measurements. Model results are shown from our standard simulation including sea salt aerosol debromination (red lines), and sensitivity simulations not including SSA debromination (blue lines) and HOBr + S(IV) reactions (purple lines). Solid lines are mean values, dotted lines are median values.**

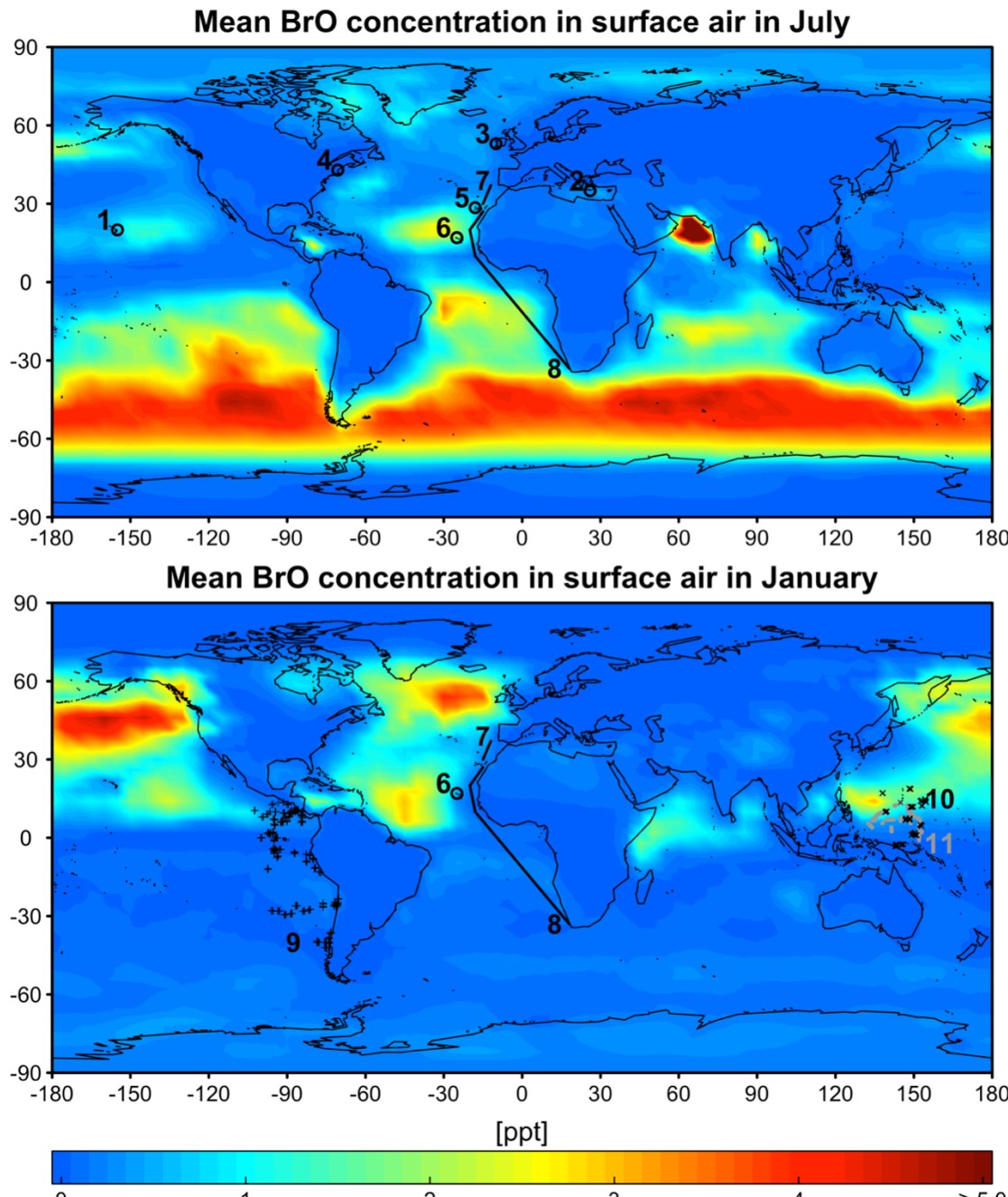

**Figure 6. GEOS-Chem mean BrO mixing ratios in surface air in July (top) and January (bottom). Locations of BrO observations in Table 1 are shown as symbols for the closest season. Open circles are ground sites and solid lines are ship tracks. Flight tracks in the marine boundary layer (< 2 km) during the TORERO (Volkamer et al., 2015; Wang et al., 2015; Dix et al., 2016), CONTRAST (Chen et al., 2016), and CAST (Le Breton et al., 2017) aircraft campaigns are shown as pluses, crosses, and dots (gray), respectively. Daytime BrO mixing ratios are about double the values shown here since BrO drops to near zero at night. Color bar saturates at 13.3 ppt near India.**

**Table 1. Daytime mixing ratios (ppt) of BrO in the marine boundary layer [a]**

| No. | Location | Time | Observed [b] | Simulated [c] | Reference [d] |
|---|---|---|---|---|---|
| | | Ground-based measurements | | | |
| 1 | Hawaii (20°N 155°W) | Sep. 1999 | < 2.0 | 1.1 | 1 |
| 2 | Crete (35°N 26°E) | Jul.–Aug. 2000 | < 0.7–1.5 | 0.48 | 1 |
| 3 | Mace Head (53°N 10°W) | Apr.–Oct. [e] | < 0.3–2.5 | 2.1 | 1, 2, 3 |
| 4 | Maine (43°N 71°W) | Jul.–Aug. 2004 | < 2.0 | 0.49 | 4 |
| 5 | Tenerife Island (29°N 17°W) | Jun.–Jul. 1997 | 3.0 | 1.0 | 1 |
| 6 | Cape Verde (17°N 25°W) | Nov. 2006–Jun. 2007 | 2.5 ± 1.9 | 1.4 | 5, 6 |
| | | Ship-based measurements | | | |
| 7 | Atlantic Ocean (30°N–37°N) | Feb. and Oct. [f] | ~ 1.0 | 0.96 | 7, 8, 9 |
| 8 | Atlantic Ocean (33°S–27°N) | Oct. 2000 | < 1.0–3.6 | 0.97 | 7 |
| | | Aircraft-based measurements | | | |
| 9 | Eastern tropical Pacific Ocean (TORERO) | Jan.–Feb. 2012 | 0.26 ± 0.15 | 0.22 | 10, 11, 12 |
| 10 | Western tropical Pacific Ocean (CONTRAST) | Jan.–Feb. 2014 | 0.63 ± 0.74 | 0.66 | 13 |
| 11 | Western tropical Pacific Ocean (CAST) | Jan.–Feb. 2014 | 0.28 ± 0.16 | 0.36 | 14 |

[a] Locations of measurements are shown in **Figure 6**.
[b] Values reported as ranges, means, and means ± standard deviations depending on availability. The symbol "<" indicates that BrO is below the corresponding detection limit.
[c] Mean values for the simulated model year of 2012. The observations are for different years. Model values are sampled at the location and time of year of the observations.
[d] 1 Sander et al. (2003), 2 Saiz-Lopez et al. (2004), 3 Saiz-Lopez et al. (2006), 4 Keene et al. (2007), 5 Read et al. (2008), 6 Mahajan et al. (2010), 7 Leser et al. (2003), 8 Martin et al. (2009), 9 Saiz-Lopez et al. (2012), 10 Volkamer et al. (2015), 11 Wang et al. (2015), 12 Dix et al., (2016); 13 Chen et al. (2016), and 14 Le Breton et al. (2017)
[e] from 1996, 1997 and 2002.
[f] from 2000 and 2007.