# Peer review of "Effect of sea-salt aerosol on tropospheric bromine chemistry"

_Atmospheric Chemistry and Physics, 2018_

## Referee Comment (RC1) · Sander (Referee) · 4 Dec 2018

Zhu et al. investigate the effect of sea-salt aerosol on tropospheric bromine chemistry. The study is very interesting and I recommend publication in ACP after considering several changes as described below.

**Specific comments**

- Abstract: *"We show, for the first time, observed levels of SSA debromination can be reproduced in a manner consistent with observed BrO concentrations."*

  A key argument of your study is that acetaldehyde is *"critical in moderating tropo-*

*spheric BrO levels"*. It should be noted, however, that this topic has already been studied in 2004 by Toyota et al. (doi 10.5194/ACP-4-1961-2004). They wrote: *"Model calculations with the new chemical scheme reveal that the oceanic emissions of acetaldehyde (CH3CHO) and alkenes (especially C3H6) are important factors for regulating reactive halogen chemistry in the MBL by promoting the conversion of Br atoms into HBr..."*

The reaction HOBr + S(IV) has also been included in previous halogen studies, e.g., Long et al. (doi 10.5194/ACP-14-3397-2014).

To support your statement about HOBr and acetaldehyde, it would be good to add a table showing the relative importance of all reactions that convert reactive bromine back to HBr in your model.

- Abstract: *"Bromine radicals influence global tropospheric chemistry by depleting ozone and OH, and by oxidizing elemental mercury, sulfur species, and volatile organic compounds."*

In the main ozone-depleting cycle (Br → BrO → HOBr → Br), the OH radical is produced, not destroyed! If you see lower OH when you switch on Br chemistry in your model, this is more likely due to indirect effects. Maybe there is less OH production because of less ozone?

I don't think that VOCs are affected much by Br. The reactions with most VOCs (e.g. alkanes) are very slow. I agree that alkenes and aldehydes can be important for Br but this doesn't necessarily imply that Br is important for them. Can you tell us what the relative contributions of OH and Br to the oxidation of aldehydes are in your model?

- Page 5, lines 8-9: *"Bry mobilized from coarse SSA is transferred to fine SSA as HBr. EF values drop further inland as Bry is remobilized from the fine SSA."*

This is very interesting! However, so far you have only presented the overall EF.

Can you calculate the EF for fine and coarse SSA individually? Then you can check if your statement is supported by the model.

- Page 4, lines 2-3: *"SSA alkalinity is depleted by uptake of HNO3 and SO2."*

  The alkalinity is not depleted by $SO_2$. This happens only after it has been oxidized to S(VI).

- Page 6, lines 10-11: *"The biases at high latitudes can be reduced by including a detail chlorine chemistry"*

  At high latitudes, tropospheric ozone depletion events occur which dominate the halogen chemistry in those regions. Unless your model includes a good description of these events, I would not attempt to compare the model results with measurements here.

**Technical Comments**

- Reactions occuring inside liquid aerosols should be called "multiphase", not "heterogeneous" (see e.g., Ravishankara, doi 10.1126/science.276.5315.1058).

- According to the IUPAC Recommendations (page 1387 of Schwartz & Warneck "Units for use in atmospheric chemistry", Pure & Appl. Chem., 67(8/9), 1377-1406, 1995, https://www.iupac.org/publications/pac/pdf/1995/pdf/6708x1377.pdf) the usage of "ppb" and "ppt" is discouraged for several reasons. Instead, "nmol/mol" and "pmol/mol" should be used for gas-phase mole fractions. I suggest to replace the obsolete units.

- Page 2, line 17 and elsewhere: *"concentrations are typically of the order of 1 ppt"*

  The physical properties "mixing ratio" and "concentration" are used as if they were identical. This is not the case! (for details, see http://www.rolf-sander.net/

res/vol1kg.pdf) Please check all occurrences of the word "concentration" in the main text and check if it should read "mixing ratio" instead.

- Data availability section: *"Data are available upon request"*

  Obtaining data via personal request can be quite unsatisfactory, especially after a couple of years when it often becomes difficult to contact the authors. I suggest to present the most important data in the supplement to this article. This could include:

  – The data files used to create the figures.
  – The reaction mechanism (KPP equation file). Did you use Standard.eqn as supplied by GEOS-Chem 12.0.0, or did you modify the mechanism?

---

## Referee Comment (RC2) · Anonymous Referee #2 · 7 Jan 2019

In this work, Zhu and co-authors presented tropospheric bromine chemistry simulated using a global chemical transport model (GEOS-Chem), and the results are compared to sea salt bromide enrichment factor measurement compilation, GOME-2 tropospheric BrO column, as well as BrO measurements from recent aircraft campaigns. This paper is very interesting and scientifically sound to merit publication in Atmospheric Chemistry and Physics once the following concerns have been addressed.

The current form of the paper is very compact. However, I also feel that some essential information is missing. My major concerns are as follows:

1. The reactive uptake of HOBr onto bromide-containng particles is critical, which is very sensitive to acidity. However, in the present form of the manuscript, neither sea salt aerosol acidity nor the titration (by HNO3, H2SO4, SO2, etc) is justified by

[Figure]

observations. This is the case for non-sea-salt aerosols and cloud droplets as well. Without proper justifications of modeled particle acidity and the acidic gases, such as HNO3, H2SO4, SO2, etc, I am not fully convinced that the modeled magnitude of sea salt debromination and bromide enrichment factors presented in the present form are necessarily for the right reason. I will get back to this later.

2. The oceanic emission of acetaldehyde (Millet et al 2010) rapidly removes Br atoms in the remote troposphere and essentially shuts off chain propagation. However, the oceanic fluxes of acetaldehyde and the modeled acetaldehyde distributions in the remote troposphere remain untested, partially due to the known measurement artifacts for acetaldehyde in clean air (please see discussions in Millet et al 2010). This weakens one major conclusion of this work, that ocean emissions of aldehyde (especially acetaldehyde) plays a key role in reconciling the large debromination source with the relatively low BrO observations in the marine boundary layer. How does the modeled acetaldehyde compare to recent observations in the remote atmosphere?

Specific comments:

3. This work does reproduce the mean bromide enrichment factors observed in ten surface sites around the globe, however the seasonality is only examined at one site. Any particular reason that the modeled seasonality is not examined against observations for the rest of the nine sites? Also, as shown in Figure 2, the modeled bromide enrichment factor shows almost the opposite annual trend, which appears to be inconsistent with the modeled alkalinity...? Anyway, this figure certainly deserves more in-depth discussions. And I would suggest adding similar plots for other sites too (depending on data availability of course).

4. The authors claim that two types of sea salt aerosols are emitted in the model: coarse and fine modes. The lifetime of sea salt aerosol depends on size. Is the modeled vertical distributions of fine and coarse sea salt aerosols supported by observations?

[Figure]

5. The author also mentioned that "Bry mobilized from coarse SSA to fine SSA as HBr", yet zero detail is given in the present form of the manuscript. Bry mobilizing from coarse SSA to fine SSA implies different enrichment factors between coarse and fine particles. Is this supported by observations? Come to think of it - is this "enrichment factor" being discussed throughout the paper weighted between coarse and fine particle?

6. The reported HOBr reactive uptake on tropospheric aerosols in the literature varies in magnitude. Roberts et al (2014) brought up a very interesting concern about the re-active uptake of HOBr on bromide-containing particles, that the widely used termolec-ular kinetic approach is strictly valid for a specific pH range, and the reactive uptake of HOBr on H2SO4-acidified particles may be substantially overestimated in current models. For the first time, Roberts et al (2014) reconciles the different reactive uptake coefficients reported from laboratory experiments. I understand that the evaluation of the new HOBr uptake framework (Roberts et al., 2014) in a global model may well be worth another paper. However, I feel this is not totally beyond the scope of the present work, given the profound impact of HOBr on the tropospheric bromine activa-tion. Therefore I would strongly suggest at least a few sensitivity tests on different pH values for sea salt and non-sea-salt aerosols. This brings me back to my very first comment: What is the pH for fresh and aged sea salt aerosols? How much acidic gases (HNO3, H2SO4) does the model predict and are these predictions supported by observations? How about non-sea-salt aerosols and cloud droplets?

7. Page 7 Line 4-5: It seems to me that the model predicts lower BrO in the free tropo-sphere in the western tropical Pacific, which happens to be a hot spot for deep convec-tion. Liang et al (2014) demonstrated that varying model deep convection strength can introduce a significant change in inorganic Bry scavenging and eventually the bromine product gas injection at the tropopause. How does the model handle convective trans-port and how may this affect the BrO (and Bry) predictions in this region?

8. Page 7 Line 5: If "oneline aerosol pH" yields better pH, why don't you use it then? This is an issue throughout, that pH is so important yet virtually no detail is given in

the present form of the paper. How much acidic gases does the model predict in the marine boundary layer and the free troposphere? How big of a difference does this "online pH calculation" make and how does this translate into the modeled BrO? To fully justify the findings (which are very interesting) in this work, I strongly suggest that the authors should get to the bottom of this. I am sure the readers will appreciate it, as it makes the paper much stronger.

Technical comments:

9. Page 1 Line 21 and Page 2 Line 2: I myself don't think "deplete" is the proper verb for OH radicals. Bromine chemistry converts HO2 into OH but also destroys O3 which itself is an OH precursor.

10. Page 1 Line 24: define BrO.

11. Page 2 Line 26: I believe Millet et al (2010) only discussed acetaldehyde, not "oxygenated organics".

12. Page 4 Line 1: please define alkalinity and remind the readers how this is derived / calculated.

13. Page 5 Line 9-10: Does Bry also "mobilize" to non-sea-salt aerosols?

14. Page 5 Line 22 and Figure 3: please separate in-cloud processing from multiphase chemistry on sulfate aerosols.

15. Page 6 Line 10-12: Please elaborate how does chlorine chemistry reduce the bias, and by how much.

16. Page 7 Line 2-4: "The CONTRAST flight campaign took more readings over a region (near Guam) with higher SSA emission and load compared with CAST, resulting in higher level of BrO" this seems to be confusing to me. As described in the present form of the manuscript, it looks like CONTRAST and CAST were pretty much at the same time in the same region, i.e. they are sort of like different subsets of the same

dataset. If more reading in one campaign leads to higher or lower values than the other, are you implying sampling bias in either or both?

17. Page 7 Line 7-9: How much HOI and I2 does the model predict in this region? And how much BrO does the model predict without iodine chemistry at this site? This is quite interesting.

18. Figure 1: please add site names onto the map.

19. I would love to see a vertical distribution of the inorganic Bry speciation plot.

References

Roberts, T. J., Jourdain, L., Griffiths, P. T., and Pirre, M.: Re-evaluating the reactive uptake of HOBr in the troposphere with implications for the marine boundary layer and volcanic plumes, Atmos. Chem. Phys., 14, 11185-11199, https://doi.org/10.5194/acp-14-11185-2014, 2014.

Liang, Q., Atlas, E., Blake, D., Dorf, M., Pfeilsticker, K., and Schauffler, S.: Convective transport of very short lived bromocarbons to the stratosphere, Atmos. Chem. Phys., 14, 5781-5792, https://doi.org/10.5194/acp-14-5781-2014, 2014.

---

## Author Comment (AC1) · 25 Apr 2019

We thank the reviewer for thorough and thoughtful comments. Please see the attached zip file for our responses, the revised manuscript, and the supplemental materials.

Please also note the supplement to this comment:
https://www.atmos-chem-phys-discuss.net/acp-2018-1239/acp-2018-1239-AC1-supplement.zip

---

## Author Comment (AC2) · 25 Apr 2019

We thank the reviewer for thorough and thoughtful comments. Please see the attached zip file under AC1 for our responses, the revised manuscript, and the supplemental materials.

---

## Author Response (AR1)

We thank the two reviewers for their thoughtful comments. Our responses are in blue. Trackable changes relative to the last manuscript are also attached at the end.

**Referee #1**
Sander (Referee)
rolf.sander@mpic.de

Zhu et al. investigate the effect of sea-salt aerosol on tropospheric bromine chemistry. The study is very interesting and I recommend publication in ACP after considering several changes as described below.

Specific comments

Abstract: "We show, for the first time, observed levels of SSA debromination can be reproduced in a manner consistent with observed BrO concentrations."
A key argument of your study is that acetaldehyde is "critical in moderating tropospheric BrO levels". It should be noted, however, that this topic has already been studied in 2004 by Toyota et al. (doi 10.5194/ACP-4-1961-2004). They wrote: "Model calculations with the new chemical scheme reveal that the oceanic emissions of acetaldehyde (CH3CHO) and alkenes (especially C3H6) are important factors for regulating reactive halogen chemistry in the MBL by promoting the conversion of Br atoms into HBr. . . "
The reaction HOBr + S(IV) has also been included in previous halogen studies, e.g., Long et al. (doi 10.5194/ACP-14-3397-2014).

Accepted. We have rephrased the words in the abstract. We now add Toyota et al. (2004) and Long et al. (2014) as references. Please see page 1 line 21.

To support your statement about HOBr and acetaldehyde, it would be good to add a table showing the relative importance of all reactions that convert reactive bromine back to HBr in your model.

Actually, we did give the relative importance of reactions converting bromine radical (Br) back to HBr in Figure 3 (term "B" at the top left corner) in the submitted manuscript. We agree with the reviewer, and add one sentence in the text to emphasize this point. Please see page 6 line 9-11.

Abstract: "Bromine radicals influence global tropospheric chemistry by depleting ozone and OH, and by oxidizing elemental mercury, sulfur species, and volatile organic compounds."
In the main ozone-depleting cycle (Br → BrO → HOBr → Br), the OH radical is produced, not destroyed! If you see lower OH when you switch on Br chemistry in your model, this is more likely due to indirect effects. Maybe there is less OH production because of less ozone?

Indeed, the less OH production is due to less ozone. We have modified this sentence. Please see page 1 line 21 and page 2 line 2.

I don't think that VOCs are affected much by Br. The reactions with most VOCs (e.g. alkanes) are very slow. I agree that alkenes and aldehydes can be important for Br but this doesn't necessarily

imply that Br is important for them. Can you tell us what the relative contributions of OH and Br to the oxidation of aldehydes are in your model?

Accepted. We have removed "VOCs" from the sentence. Please see page 1 line 21 and page 2 line 2. In the model, OH accounts for 96.5% (Br 3.5%) of the total oxidation of HCHO within the troposphere. For acetaldehyde, OH accounts for 92.0% (Br 8.0%) of the total oxidation within the troposphere.

Page 5, lines 8-9: "Bry mobilized from coarse SSA is transferred to fine SSA as HBr. EF values drop further inland as Bry is remobilized from the fine SSA." This is very interesting! However, so far you have only presented the overall EF. Can you calculate the EF for fine and coarse SSA individually? Then you can check if your statement is supported by the model.

Accepted. We now calculate and discuss the *EF* values for fine and coarse SSA separately. Please see page 14 for the revised Figure 1, and page 5 line 17-23 for discussion.

Page 4, lines 2-3: "SSA alkalinity is depleted by uptake of HNO3 and SO2." The alkalinity is not depleted by SO2. This happens only after it has been oxidized to S(VI).

Accepted. We have rephrased this. Please see page 4 line 14-15.

Page 6, lines 10-11: "The biases at high latitudes can be reduced by including a detail chlorine chemistry" At high latitudes, tropospheric ozone depletion events occur which dominate the halogen chemistry in those regions. Unless your model includes a good description of these events, I would not attempt to compare the model results with measurements here.

Accepted. We have included the comprehensive tropospheric chlorine chemistry from Wang et al. (2019) with some additional modifications, and have updated the manuscript accordingly. We also add one sentence to clarify the scope of the manuscript, please see page 3 line 20-22.

Technical Comments

Reactions occuring inside liquid aerosols should be called "multiphase", not "heterogeneous" (see e.g., Ravishankara, doi 10.1126/science.276.5315.1058).

We appreciate the reviewer's comment, but "heterogeneous" is a standard terminology in our field and people have used it in many previous studies. So, we prefer to use it in the manuscript.

According to the IUPAC Recommendations (page 1387 of Schwartz & War- neck "Units for use in atmospheric chemistry", Pure & Appl. Chem., 67(8/9), 1377-1406, 1995, https://www.iupac.org/publications/pac/pdf/1995/pdf/ 6708x1377.pdf) the usage of "ppb" and "ppt" is discouraged for several reasons. Instead, "nmol/mol" and "pmol/mol" should be used for gas-phase mole fractions. I suggest to replace the obsolete units.

We appreciate the reviewer's comment, but the term "ppb/ppt" is widely used in our field. We accept this comment by defining ppt as at its first appearance. Please see page 2 line 17.

Page 2, line 17 and elsewhere: "concentrations are typically of the order of 1 ppt" The physical properties "mixing ratio" and "concentration" are used as if they were identical. This is not the case! (for details, see http://www.rolf-sander.net/res/vol1kg.pdf) Please check all occurrences of the word "concentration" in the main text and check if it should read "mixing ratio" instead.

Accepted. We have replaced "concentration" with "mixing ratio" in multiple places in the text.

Data availability section: "Data are available upon request". Obtaining data via personal request can be quite unsatisfactory, especially after a couple of years when it often becomes difficult to contact the authors. I suggest to present the most important data in the supplement to this article. This could include:
–        The data files used to create the figures.
–        The reaction mechanism (KPP equation file). Did you use Standard.eqn as supplied by GEOS-Chem 12.0.0, or did you modify the mechanism?

Accepted. We have uploaded the KPP files and data used to create figures, and have modified the Data availability section accordingly. Please see page 8 line 29-30.

**Anonymous Referee #2**

In this work, Zhu and co-authors presented tropospheric bromine chemistry simulated using a global chemical transport model (GEOS-Chem), and the results are compared to sea salt bromide enrichment factor measurement compilation, GOME-2 tropospheric BrO column, as well as BrO measurements from recent aircraft campaigns. This paper is very interesting and scientifically sound to merit publication in Atmospheric Chemistry and Physics once the following concerns have been addressed.

The current form of the paper is very compact. However, I also feel that some essential information is missing. My major concerns are as follows:

1. The reactive uptake of HOBr onto bromide-containng particles is critical, which is very sensitive to acidity. However, in the present form of the manuscript, neither sea salt aerosol acidity nor the titration (by HNO3, H2SO4, SO2, etc) is justified by observations. This is the case for non-sea-salt aerosols and cloud droplets as well. Without proper justifications of modeled particle acidity and the acidic gases, such as HNO3, H2SO4, SO2, etc, I am not fully convinced that the modeled magnitude of sea salt debromination and bromide enrichment factors presented in the present form are necessarily for the right reason. I will get back to this later.

We agree with the reviewer that acidity is critical for the reactive uptake of HOBr in cloud and on sea salt aerosol. We have added one sentence to compare model cloud pH with observations reported in literatures. Please see page 4 line 15-17. We have also added a figure showing sea salt aerosol alkalinity in the Supplemental Materials (Figure S1).

2. The oceanic emission of acetaldehyde (Millet et al 2010) rapidly removes Br atoms in the remote troposphere and essentially shuts off chain propagation. However, the oceanic fluxes of acetaldehyde and the modeled acetaldehyde distributions in the remote troposphere remain untested, partially due to the known measurement artifacts for acetaldehyde in clean air (please see discussions in Millet et al 2010). This weakens one major conclusion of this work, that ocean emissions of aldehyde (especially acetaldehyde) plays a key role in reconciling the large debromination source with the relatively low BrO observations in the marine boundary layer. How does the modeled acetaldehyde compare to recent observations in the remote atmosphere?

Based on a recent study, GEOS-Chem acetaldehyde is 10% lower compared with observations over remote MBL from the ATom campaigns. By assuming such bias is uniform in the model, global tropospheric Br radical to HBr rate would be increased only by 4%. Please see page 6 line 12-14.

Specific comments:

3. This work does reproduce the mean bromide enrichment factors observed in ten surface sites around the globe, however the seasonality is only examined at one site. Any particular reason that the modeled seasonality is not examined against observations for the rest of the nine sites? Also, as shown in Figure 2, the modeled bromide enrichment factor shows almost the opposite annual trend, which appears to be inconsistent with the modeled alkalinity...? Anyway, this figure

certainly deserves more in-depth discussions. And I would suggest adding similar plots for other sites too (depending on data availability of course).

We appreciate the reviewer's comment, however, Gape Grim is the only site with seasonal *EF* observations we could find. We have rewritten this part to clarify. Please see page 5 line 17-27.

4. The authors claim that two types of sea salt aerosols are emitted in the model: coarse and fine modes. The lifetime of sea salt aerosol depends on size. Is the modeled vertical distributions of fine and coarse sea salt aerosols supported by observations?

GEOS-Chem sea salt aerosol simulation is based on Jaeglé et al. (2011) and has been evaluated with surface, cruise, satellite, and AERONET data. We think validation of sea salt aerosol profiles is beyond the scope of this study. However, we accept the reviewer's comment by adding "showed that it could reproduce successfully SSA observations over the oceans from ship cruises and coastal/island stations, as well as observations of aerosol optical depth (AOD) from the AERONET network and the MODIS satellite instrument" in the text. Please see page 4 line 8-10.

5. The author also mentioned that "Bry mobilized from coarse SSA to fine SSA as HBr", yet zero detail is given in the present form of the manuscript. Bry mobilizing from coarse SSA to fine SSA implies different enrichment factors between coarse and fine particles. Is this supported by observations? Come to think of it is this "enrichment factor" being discussed throughout the paper weighted between coarse and fine particle?

We calculated *EF* values using the total mass of bromide in SSA and the total mass of dry SSA, from both fine and coarse SSA. We state this in the text, please see page 5 line 4. We have accepted the reviewer's comment by calculating and discussing the *EF* values for fine and coarse SSA separately. Please see page 14 for the revised Figure 1, and page 5 line 17-23 for discussion.

6. The reported HOBr reactive uptake on tropospheric aerosols in the literature varies in magnitude. Roberts et al (2014) brought up a very interesting concern about the re-active uptake of HOBr on bromide-containing particles, that the widely used termolecular kinetic approach is strictly valid for a specific pH range, and the reactive uptake of HOBr on H2SO4-acidified particles may be substantially overestimated in current models. For the first time, Roberts et al (2014) reconciles the different reactive uptake coefficients reported from laboratory experiments. I understand that the evaluation of the new HOBr uptake framework (Roberts et al., 2014) in a global model may well be worth another paper. However, I feel this is not totally beyond the scope of the present work, given the profound impact of HOBr on the tropospheric bromine activation. Therefore I would strongly suggest at least a few sensitivity tests on different pH values for sea salt and non-sea-salt aerosols. This brings me back to my very first comment: What is the pH for fresh and aged sea salt aerosols? How much acidic gases (HNO3, H2SO4) does the model predict and are these predictions supported by observations? How about non-sea-salt aerosols and cloud droplets?

We appreciate the reviewer's comment on possible overestimation of reactive uptake of HOBr in the model. However, our simulated BrO is generally toward the low end of the observed range of BrO (Figure 5 and Table 1). This implies that slowing down HOBr + Br$^-$ may not help improve

the simulation. We have accepted the reviewer's comment by citing Roberts et al (2014) as a possible future direction. Please see page 8 line 20.

"SSA alkalinity is emitted at a ratio of 0.07 equivalents per kilogram of dry SSA, and is depleted by strong acids following Alexander et al. (2005)". We have added this sentence in the text to address the reviewer's comment. Please see page 4, line 14-15. Currently, the model only considers $HOBr + Br^-$ reactions in cloud droplets, ice crystals, SSA, and sulfate aerosol. We have modified one sentence in the text to clarify this. Please see page 3 line 11.

7. Page 7 Line 4-5: It seems to me that the model predicts lower BrO in the free troposphere in the western tropical Pacific, which happens to be a hot spot for deep convection. Liang et al (2014) demonstrated that varying model deep convection strength can introduce a significant change in inorganic Bry scavenging and eventually the bromine product gas injection at the tropopause. How does the model handle convective transport and how may this affect the BrO (and Bry) predictions in this region?

Such bias is no longer the case. Please see page 7 line 19-21.

8. Page 7 Line 5: If "oneline aerosol pH" yields better pH, why don't you use it then? This is an issue throughout, that pH is so important yet virtually no detail is given in the present form of the paper. How much acidic gases does the model predict in the marine boundary layer and the free troposphere? How big of a difference does this "online pH calculation" make and how does this translate into the modeled BrO? To fully justify the findings (which are very interesting) in this work, I strongly suggest that the authors should get to the bottom of this. I am sure the readers will appreciate it, as it makes the paper much stronger.

Accepted. We have included the comprehensive tropospheric chlorine chemistry from Wang et al. (2019) with some additional modifications, and updated the manuscript accordingly.

Technical comments:
9. Page 1 Line 21 and Page 2 Line 2: I myself don't think "deplete" is the proper verb for OH radicals. Bromine chemistry converts HO2 into OH but also destroys O3 which itself is an OH precursor.

Accepted. We have modified this sentence. Please see page 1 line 21 and page 2 line 2.

10. Page 1 Line 24: define BrO.

Accepted. We have defined BrO as "bromine monoxide". Please see page 1 line 24.

11. Page 2 Line 26: I believe Millet et al (2010) only discussed acetaldehyde, not "oxygenated organics".

Accepted. We have modified the sentence. Please see page 2 line 27.

12. Page 4 Line 1: please define alkalinity and remind the readers how this is derived/calculated.

Accepted. We have rewritten this sentence. Please see page 4 line 14-15.

13. Page 5 Line 9-10: Does Bry also "mobilize" to non-sea-salt aerosols?

We now calculate and discuss the *EF* values for fine and coarse SSA separately. Please see page 14 for the revised Figure 1, and page 5 line 17-23 for discussion.

14. Page 5 Line 22 and Figure 3: please separate in-cloud processing from multiphase chemistry on sulfate aerosols.

We appreciate the reviewer's comment. Unfortunately, our current code doesn't allow separation of in-cloud processing between heterogenous chemistry on sulfate aerosols.

15. Page 6 Line 10-12: Please elaborate how does chlorine chemistry reduce the bias, and by how much.

We have updated the manuscript using the comprehensive tropospheric chlorine chemistry following Wang et al. (2019). Therefore, this part has been removed in the revised manuscript.

16. Page 7 Line 2-4: "The CONTRAST flight campaign took more readings over a region (near Guam) with higher SSA emission and load compared with CAST, resulting in higher level of BrO" this seems to be confusing to me. As described in the present form of the manuscript, it looks like CONTRAST and CAST were pretty much at the same time in the same region, i.e. they are sort of like different subsets of the same dataset. If more reading in one campaign leads to higher or lower values than the other, are you implying sampling bias in either or both?

CONTRAST took more observations near Guam where BrO level is higher due to higher SSA emissions. We have rewritten this sentence. Please see page 7 line 23-24. We have also modified Figure 6 (page 19) by showing aircraft observations for the closest season.

17. Page 7 Line 7-9: How much HOI and I2 does the model predict in this region? And how much BrO does the model predict without iodine chemistry at this site? This is quite interesting.

Accepted. Such hotspot becomes weaker in the revised manuscript. We have rewritten this sentence. Please see page 7 line 25-27.

18. Figure 1: please add site names onto the map.

Accepted. Please see page 14 line 6-7.

19. I would love to see a vertical distribution of the inorganic Bry speciation plot. Can we do this as an update to the corresponding plot from Schmidt2016?

Accepted. Please see Figure S3 and S4 in the Supplemental Materials.

[revised manuscript text omitted]